# Virus-Like Particles Based on the Novel Goose Parvovirus (NGPV) VP2 Protein Protect Ducks against NGPV Challenge

**DOI:** 10.3390/vaccines11121768

**Published:** 2023-11-27

**Authors:** Yu Shang, Yao Ma, Sheng Tang, Xing Chen, Helong Feng, Li Li, Hongcai Wang, Zhe Zeng, Lun Yao, Tengfei Zhang, Chi Zeng, Qingping Luo, Guoyuan Wen

**Affiliations:** 1Key Laboratory of Prevention and Control Agents for Animal Bacteriosis, Ministry of Agriculture and Rural Affairs, Hubei Provincial Key Laboratory of Animal Pathogenic Microbiology, Institute of Animal Husbandry and Veterinary, Hubei Academy of Agricultural Sciences, Wuhan 430064, China; suppershangyu@126.com (Y.S.); myjay1218@163.com (Y.M.); zengzhe@hbaas.com (Z.Z.); tfzhang23@163.com (T.Z.); 2School of Life Science and Technology, Wuhan Polytechnic University, Wuhan 430023, China; czeng@whpu.edu.cn; 3Institute of Animal Husbandry and Veterinary, Wuhan Academy of Agricultural Sciences, Wuhan 430071, China; ligang1997@126.com; 4Hubei Hongshan Laboratory, the People’s Government of Hubei Province, Wuhan 430070, China

**Keywords:** novel goose parvovirus, VP2 protein, virus-like particles, immunogenicity

## Abstract

Novel goose parvovirus (NGPV), a genetic variant of goose parvovirus, has been spreading throughout China since 2015 and mainly infects ducklings with the symptoms of growth retardation, beak atrophy, and protruding tongue, leading to huge economic losses every year. A safe and effective vaccine is urgently needed to control NGPV infection. In this study, virus-like particles (VLPs) of NPGV were assembled and evaluated for their immunogenicity. The VP2 protein of NGPV was expressed in *Spodoptera frugiperda* insect cells using baculovirus as vector. The VP2 protein was efficiently expressed in the nucleus of insect cells, and the particles with a circular or hexagonal shape and a diameter of approximately 30 nm, similar to the NGPV virion, were observed using transmission electron microscopy (TEM). The purified particles were confirmed to be composed of VP2 using western blot and TEM, indicating that the VLPs of NGPV were successfully assembled. Furthermore, the immunogenicity of the VLPs of NGPV was evaluated in Cherry Valley ducks. The level of NGPV serum antibodies increased significantly at 1–4 weeks post-immunization. No clinical symptoms or deaths of ducks occurred in all groups after being challenged with NGPV at 4 weeks post-immunization. There was no viral shedding in the immunized group. However, viral shedding was detected at 3–7 days post-challenge in the non-immunized group. Moreover, VLPs can protect ducks from histopathological lesions caused by NGPV and significantly reduce viral load in tissue at 5 days post-challenge. Based on these findings, NGPV VLPs are promising candidates for vaccines against NGPV.

## 1. Introduction

Short beak and dwarfism syndrome (SBDS), characterized by growth retardation, smaller beak, and tarsus, is a widely spreading infectious disease in ducklings caused by novel goose parvovirus (NGPV) with high morbidity and a low mortality rate [1,2]. In 1971–1972, a large number of mule ducks in southwest France showed SBDS, and the molecular identification proved that the parvovirus isolates obtained from this case belonged to a distinct lineage of a goose parvovirus-related group of waterfowl parvoviruses in 2009 [3]. In 1989, a new acute duckling disease with high morbidity and mortality outbroke in Taiwan, the surviving ducks showed growth retardation and shorter beaks, and the culprit turned out to be NGPV [4]. In 2015, the NGPV-related disease in Cherry Valley ducks characterized by growth retardation and short beaks was first reported in the Shandong province of China [5]. Since 2019, SBDS has been observed in some mule and Pekin duck farms in Egypt and in Pekin duck farms in Poland [6,7]. In some cases, the appearance of mixed infection by NGPV and other pathogens seriously affected the emergence rate of ducks and caused huge economic losses to the waterfowl breeding industry [8].

NGPV belongs to the species *Anseriform dependoparvovirus* 1 of the genus *Dependoparvovirus* and family *Parvovirinaehas* [9], which can proliferate in duck embryo fibroblasts but not in goose embryo fibroblasts [10]. Its genome is single-stranded DNA with a size of approximately 5.1 kb containing two open reading frames (ORFs) of the left and right ORF. The left ORF encodes nonstructural proteins, including NS1 and NS2, and the right ORF encodes structural proteins, including VP1, VP2, and VP3. The initiation codons for VP1, VP2, and VP3 are at different sites but share the same stop codons [5].

Currently, there is no effective vaccine for the prevention and control of NGPV. Anti-NGPV egg yolk antibodies are used clinically for controlling NGPV but pose potential biosafety risks [8]. It is crucial to develop a safe and effective vaccine against NGPV. Virus-like particles (VLPs) are assembled from one or more structural proteins of the virus and have a similar structure to natural viruses, exhibiting high immunogenicity and safety [11,12,13]. In this study, the VLPs composed of the NGPV VP2 protein were produced via the baculovirus expression vector system (BEVS). The immunogenicity, safety, and efficiency of VLPs were evaluated.

## 2. Material and Methods

### 2.1. Virus, Plasmid Vector and Cells

*Spodoptera frugiperda* (Sf9) insect cells were cultured in Serum Free Medium (CELLiGENT, Hamilton, New Zealand) at 27 °C in a constant temperature incubator. The NGPV Yich strain was isolated from a clinically diseased duck using 12–14 day-old duck embryo. The virus had a titer of 10^4.8^ EID_50_/mL and was maintained in the laboratory. The partial genome of Yich strain was submitted to the National Center for Biotechnology Information (https://www.ncbi.nlm.nih.gov/) and Genbank number was MZ359667.1. The “Bac-to-Bac” baculovirus expression system was kindly provided by Zhihong Hu, Professor of Wuhan Institute of Virology, Chinese Academy of Sciences. In the baculovirus expression system, *Escherichia coli* DH10Bac cells containing a baculovirus genome bacmid (AcBac-∆cc) [14,15] and a helper plasmid encoding the Tn7 transposition enzyme donor were cultured on Luria-Bertani medium with kanamycin and tetracycline, and a donor vector pFastBac-Dual was amplified in *E. coli* DH5α (Vazyme Biotech, Nanjing, China).

### 2.2. Construction and Rescue of Recombinant Baculovirus

The construction strategy of the recombinant baculovirus genome was shown in Figure 1A. First, the VP2 gene was amplified using polymerase chain reaction (PCR) using NGPV genomic DNA as a template and primer pairs VP2-F3 and VP2-R2 (Table 1). A fragment containing Flag tag codon sequences was synthesized via primer self-extension using primer pairs FH-F1 and FH-R1 and then fused with the VP2 gene fragment by overlapping PCR to form a chimeric VP2 expression box (tVP2). The tVP2 fragment and pFastBac-Dual vector were digested using *Xho* I and *Sph* I restriction endonucleases (Takara Bio, Shiga, Japan) and linked together using T4 DNA ligase (Takara) to construct the recombinant plasmid pFB-tVP2. pFB-tVP2 was transformed into competent cells of *E. coli* DH5α for amplification then extracted and purified using the EasyPure Plasmid MiniPrep Kit (TransGen Biotech, Beijing, China). Then, pFB-tVP2 plasmid was transformed into the competent cells of *E. coli* BH10Bac containing AcBac-∆cc bacmid and helper plasmid to generate the recombinant bacmid pAc-tVP2. A single colony was selected for amplification and identification using PCR. The pAc-tVP2 plasmid was extracted and purified using a QIAGEN Plasmid Mini Kit (QIAGEN, Germantown, MD, USA).

Then, 5 μg DNA of pAc-tVP2 was transfected into Sf9 insect cells using Cellfection^®^ II Reagent (Gibco, Thermo Fisher Scientific, Waltham, MA, USA). Cytopathic effects were observed at 48–120 h post-transfection (hpt). The culture supernatant was collected at 120 hpt and used to infect Sf9 cells to obtain a high-titer recombinant baculovirus, rAc-tVP2. The same method was applied to generate the control baculoviruses derived from the AcBac-∆cc bacmid, named rAcBac. The titers of both viruses were determined using an endpoint dilution assay.

### 2.3. Indirect Immunofluorescence Assay Analysis

Sf9 cells were seeded into six well plates. When the cells covered 70–80% of the area, they were infected with virus rAc-tVP2 or rAcBac at a multiplicity of infection (MOI) of 5 and cultured at 27 °C for 72 h. The cells were fixed with 4% paraformaldehyde for 30 min at 27 °C, permeabilized with 0.5% Triton X-100, and blocked with 3% bovine serum albumin. The cells were then washed with phosphate-buffered saline (PBS) containing 0.1% Tween 20 (PBST) and incubated with mouse antiserum against the Flag tag (diluted 1:1500 in PBST, BOSTER, Wuhan, China) at 27 °C for 2 h. After washing thrice with PBST, the cells were stained with fluorescein isothiocyanate isomer-labeled goat anti-mouse IgG (H + L) (diluted 1:300 in PBST; Beyotime, Shanghai, China) at 27 °C for 1 h. The cells were re-washed three times prior to staining the nucleus with 4′,6-diamidino-2-phenylindole (DAPI, Beyotime, Shanghai, China) for 5 min. Finally, the plates were examined using a confocal laser microscope (ZEISS, Oberkochen, Germany) after washing thrice with PBST.

### 2.4. SDS-PAGE and Western Blotting Analysis

The Sf9 cells were infected with virus rAc-tVP2 or rAcBac at an MOI of 5.0 and cultured at 27 °C for 96 h. The cells were collected via centrifugation, resuspended in PBS (pH 7.4), and lysed by ultrasonic crushing. The cell debris and supernatant were separated via centrifugation at 4 °C with 12,000 rpm for 30 min, and the total protein was boiled for 10 min in sodium dodecyl sulfate-polyacrylamide gel electrophoresis (SDS-PAGE) sample loading buffer (Beyotime, Shanghai, China) and subjected to SDS-PAGE. Samples on 10% SDS-polyacrylamide gels were stained with Coomassie Brilliant Blue G-250 or transferred onto nitrocellulose membranes (Merck Millipore, Burlington, MA, USA) using a Mini PROTEAN ^®^ Tetra Cell (Bio-Rad, Hercules, CA, USA). Mouse antiserum against the Flag tag (diluted 1:2000 in PBST, BOSTER, Wuhan, China) was used as the primary antibody and horseradish peroxidase-labeled goat anti-mouse IgG (diluted 1:2000 in PBST, BOSTER, Wuhan, China) as the secondary antibody for western blot analysis. Antibody binding was detected using the enhanced chemiluminescence system (Merck, Darmstadt, Germany).

### 2.5. Transmission Electron Microscopy

The Sf9 cells infected with the recombinant baculoviruses of rAc-tVP2 for 72 h were collected and fixed in 2.5% glutaraldehyde (pH 7.4) for 2 h and post-fixed in 1% osmic acid at 4 °C for 2 h. Subsequently, the samples were embedded in Epon-Araldite resin and cut into ultrathin sections. This was followed by counterstaining with 3% uranyl acetate and 2.7% lead citrate. Ultrastructural analysis was performed using an HT7800 transmission electron microscope (TEM) (Hitachi, Tokyo, Japan).

### 2.6. VLP Purification

Sf9 cells infected with the recombinant baculoviruses of rAc-tVP2 for 72 h were collected by centrifugation at 1000 rpm for 5 min at room temperature and resuspended in 10 mM Tris-HCl (pH 8.0). The ultrasonic crushing method was used to break the cells, and the cell debris was separated from the supernatant via centrifugation at 12,000 rpm for 30 min. The VLPs in the supernatant were pelleted via ultracentrifugation (100,000× *g* for 60 min). The pellet was suspended in 8 mL of extraction buffer (5 mM magnesium chloride, 5 mM calcium chloride, 150 mM sodium chloride, 20 mM 4-(2-hydroxyethyl)-1-piperazineethanesulfonic acid, and 0.01% TritonX-100), sonicated for 1 min at a frequency of 28 KHz, and centrifuged at 12,000 rpm for 30 min at 4 °C. The supernatant was transferred to a centrifuge tube lined with 40% sucrose and centrifuged at 4 °C for 3 h (30,000 rpm, SW32Ti). The precipitate was resuspended in an extraction buffer and centrifuged using a non-continuous cesium chloride (1.15–1.35 g/cm^3^) gradient for 18 h at 10 °C (40,000 rpm, SW32Ti). The purified sample was collected, diluted with extraction buffer, centrifuged at 10 °C for 3 h (40,000 rpm, SW32Ti), and the precipitate was resuspended in 10 mM Tris-HCl (pH 8.0). The effect of the purification was analyzed using SDS-PAGE and western blot.

### 2.7. VLP Identification

The particle size distributions of the purified samples were determined using Zetasizer Nano ZS90 (Malvern Panalytical, Malvern, UK). Simultaneously, a small amount of the purified sample was dropped on the copper mesh with the supporting membrane, and the excess liquid was blotted out to air for drying. The samples were stained with 2% phosphotungstic acid solution for 3–5 min and then observed using TEM.

### 2.8. Animal Experiments

A total of 60 ducks aged 1 day from a commercial duck hatchery in Hubei province of China were divided randomly into two groups, VLPs (*n* = 20) and PBS (*n* = 40). Ducks in the VLPs group were immunized twice with the VLPs vaccine via intramuscular injection into the leg at 1 and 14 days old, respectively. A mixture of purified VLPs mixed with a water-in-mineral oil emulsion (also known as Montanide ISA71VG^®^, SEPPIC, Paris, France) at a volume ratio of 1:1.5 was used as a vaccine for primary and booster immunization. The concentration of VLPs was determined using the BCA Protein Assay kit (Beyotime, Shanghai, China), and the immune dose for each duck was 50 µg in a volume of 200 µL. Synchronously, the ducks in the PBS group received immunizations with 200 µL sterile PBS twice. Leg vein blood samples were collected at 1, 2, 3, and 4 weeks post-primary immunization, and the sera were separated. Serum antibody levels were detected using indirect enzyme-linked immunosorbent assay (ELISA) [16]. Two weeks post-booster immunization, the PBS group was divided into two groups: one group of ducks (*n* = 20) was challenged with NGPV Yich stain at a dose of 10^4.0^ egg infectious dose at 50% (EID_50_) (2 mL/duck) via intramuscular injection (NGPV group); the other group (*n* = 20) was challenged with 2 mL/duck sterile PBS (PBS group). Ducks in the VLPs group were challenged with NGPV Yich stain at a dose of 10^4.0^ EID_50_ (2 mL/duck) via intramuscular injection and named the VLPs + NGPV group. Ten cloacal swabs were collected randomly from each group at 1, 3, 5, and 7 days post-challenge (dpc). Three ducks from each group were randomly selected and sacrificed at 5 dpc. The livers and spleens were collected for virus titration and histopathological analysis. For histological analysis, the tissues were fixed in 4% paraformaldehyde, embedded in paraffin, sectioned, stained with hematoxylin and eosin, and analyzed microscopically.

### 2.9. Quantitative Real-Time PCR

NGPV loads in the liver, spleen, and cloacal swabs were quantified using quantitative real-time PCR (qPCR). The primer pairs targeting the NGPV NS helicase gene SF3 domain were designed as follows: forward, 5′-TACAATGGAACACAGAATWCC-3′ and reverse, 5′-TCAGACACAACAGGAACWA-3′. Viral DNA extracted from tissue or cloacal swab samples using a viral DNA kit (Omega Bio-Tek, Norcross, GA, USA) was used as the template for qPCR. The qPCR reaction system was 20 µL in volume, containing 0.5 µL forward primer, 0.5 µL reverse primer, 10 µL AceQ Universal SYBR qPCR Master Mix (Vazyme Inc., Nanjing, China), 2.0 µL of the template DNA, and 7.0 µL ddH_2_O. The cycling conditions were 95 °C for 10 min, followed by 40 cycles of 95 °C for 10 s and 60 °C for 30 s.

### 2.10. Statistical Analysis

The GraphPad Prism 8.0 was used to analyze data. All data are expressed as the mean ± standard deviation of each group. The statistical significance of the data was calculated using a one-factor analysis of variance between the experimental groups. The significance was divided into three levels: 0.05 level (*p* < 0.05) represented by *, 0.01 level (*p* < 0.01) represented by **, and 0.001 level (*p* < 0.001) represented by ***.

## 3. Results

### 3.1. Construction and Rescue of Recombinant Baculovirus

The VP2 gene amplified from the NGPV Yich strain was fused with a Flag tag fragment to construct a tVP2 fragment, and the target fragment was 1381 bp in length (Figure 1B, line 1). The tVP2 gene was inserted downstream of the p10 promoter of the pFastBac-Dual vector to construct the donor plasmid pFB-tVP2. The pFB-tVP2 plasmid was verified to be constructed correctly using restriction endonuclease digestion and sequencing (Figure 1B, line 2). After transposition, the recombinant bacmid pAcBac-tVP2 was further confirmed using blue-white screening and PCR using three pairs of primers (Figure 1B, lines 3–5), indicating that the recombinant baculovirus genome was correctly constructed.

Transfection and infection assays were performed to rescue the recombinant baculovirus. The supernatant fluid of Sf9 cells transfected with pAcBac-tVP2 or pAcBac-∆cc was collected and used to infect fresh cells. Almost all cells exhibited the typical cytopathic effects of cell swelling and hypertrophied nuclei at 96 h post-infection (hpi) (Figure 1C), indicating that AcBac-tVP2 could produce infectious viruses.

### 3.2. Expression and Identification of tVP2 Protein in Insect Cells

To detect VP2 protein expression, Sf9 cells infected with rAcBac-tVP2 or rAcBac for 72 h were subjected to an indirect immunofluorescence assay procedure. The results showed that the green fluorescence occurred in the cells infected with rAcBac-tVP2 but not in the cells infected with rAcBac. Green fluorescence was visible in the nuclei of infected cells, and partial green fluorescence clustered together (Figure 2A). The whole-cell, supernatant, and precipitate samples of cell lysis were collected from rAcBac-tVP2 or rAcBac-infected cells after ultrasonic crushing and were subjected to SDS-PAGE and western blot analysis. The results showed that the tVP2 protein, with an approximate molecular weight of 70 kDa, was detected in the whole-cell sample and supernatant samples (Figure 2B,C). These data indicated that the tVP2 protein was expressed in Sf9 cells.

### 3.3. Observation, Purification and Verification of VLPs

To observe the assembly of VLPs in the cells, Sf9 cells infected with rAcBac-VP2 were collected at 72 hpi and prepared as ultrathin sections for ultrastructural analysis using TEM. The infected cells were swollen, the nuclei were hypertrophied, and two types of particles with rod and globular shapes were observed in the nucleus. The rod-shaped particles with a size of 50 × 300 nm fit the size and shape of baculovirus nuclear capsids. The globular particles with a diameter of 30 nm were approximately the same size as those of NGPV, suggesting that the VP2 protein may be assembled into VLPs in the nucleus (Figure 2D,E).

To further verify the assembly of VLPs, the particles were purified from infected Sf9 cells. Purified particles were assayed using SDS-PAGE and western blot. The results showed that a single band emerged at the level corresponding to 70 kDa on the polyacrylamide gel after staining with Coomassie Brilliant Blue G-250, and a specific band at the same position was detected in western blot assays using a murine anti-Flag polyclonal antibody, indicating that the purified particles were composed of the VP2 protein (Figure 3A,B).

The purified VLPs were adsorbed onto a copper mesh, stained, and observed using TEM. Hexagonal or round particles with a diameter of approximately 30 nm were observed (Figure 3C), which were similar in shape and size to native NGPV. The dimensions of the purified particles were further determined using a Zetasizer Nano ZS90, which showed that the particles were approximately 30 nm in size (Figure 3D). These data were consistent with those for wild-type viruses [17], confirming that VP2 protein-assembled VLPs in Sf9 cells were similar to native viruses in shape and size.

### 3.4. Serum Antibodies Induced by VLP Vaccine

The prepared VLP vaccine was used to immunize the ducks via intramuscular injection in the leg, and sera were collected at different time points (Figure 4A). None of the immunized ducks experienced adverse reactions. The indirect ELISA was used to detect the VP2 antibody levels. The results showed that, compared with the PBS group, the antibody level of the VLPs group was significantly increased at 7, 14, 21, and 28 days after the primary immunization (Figure 4B), indicating that VLPs could induce antibodies against NGPV in ducks.

### 3.5. Protective Efficacy of VLPs against NGPV Challenge

To evaluate the protective efficacy of the VLPs, immunized ducks were challenged with the NGPV Yich strain. No clinical signs were observed in any group during the challenge test. The liver and spleen samples of ducks from each group were taken at 5 dpc, and histopathological analyses were performed. As shown in Figure 4C, no histopathological lesions were observed in the liver or spleen tissues of the VLPs + NGPV and PBS groups. Moderate histopathological lesions were observed in the liver (hepatocyte necrosis and fatty degeneration) and spleen tissues (unclear boundaries between the medulla and cortex, as well as lymphocytopenia) of ducks in the NGPV group. Furthermore, we analyzed the viral loads in the liver and spleen at 3, 5, and 7 dpc using qPCR. No virus was detected in the liver and spleen of the VLPs + NGPV group. The viruses were detected from the samples taken at 3, 5 and 7 dpc in the NGPV group (Figure 4D,E).

Finally, we investigated whether the VLP vaccines could inhibit viral shedding after the challenge. As presented in Table 2, no virus was detected in the cloacal swab samples from the VLPs + NGPV groups. In the NGPV group, the positive samples were first detected at 3 dpc, and the positive rate was the highest at 5 dpc (40%). These results demonstrated that the VLP vaccine could protect immunized ducks against NGPV infection by reducing histopathological lesions, virus replication in the liver and spleen, and viral shedding.

## 4. Discussion

VLPs are non-replicative and non-infectious particles that are morphologically similar to native viruses, lack viral genetic material, and are formed by structural proteins with inherent self-assembly capabilities [18]. They are used as a gene delivery vector to develop therapeutic agents and as vaccines to control infectious diseases [19,20,21]. Compared with attenuated virus-derived vaccines, VLPs have a higher safety profile [22,23]. Their interaction with the innate immune system can further promote the adaptive immune response [24]. Currently, many commercial vaccines are based on VLPs, such as Sci-B-Vac (hepatitis B virus), Gardasil (human papillomavirus), Mosquirix (malaria), and Gardasil9 (human papillomavirus) [25].

VLPs can be produced in various cellular systems, including bacteria, yeast, insects, plants, and mammals [26]. The BEVS is a powerful eukaryotic expression system capable of extremely high levels of protein production in insect cells combined with complex eukaryotic post-translational protein modifications, which may be important for the correct self-assembly and release of some VLPs [27]. Therefore, BEVS represents an attractive and valuable platform for VLP production.

NGPV is a variant of goose parvovirus that can infect young ducks and cause an infectious disease characterized by short duck beaks and dwarfism syndrome. In 2015, the disease broke out in China’s duck herds, seriously affected the emergence rate of ducks, and caused huge economic losses to China’s breeding industry [28]. Vaccination is the most cost-effective way to prevent disease. However, it takes a long time to attenuate the virus or inactivate wild-type viruses in the preparation of attenuated or inactivated vaccines [29]. Moreover, virulence reversion and incomplete inactivation are important factors affecting the safety of live and inactivated vaccines, respectively. In this study, we used BEVS to prepare NGPV VLPs, hoping to develop a safe and efficient vaccine against NGPV.

The VP2 protein is most commonly used to develop VLPs of GPV and NGPV. In the previous research, the VP2 proteins of GPV or NGPV were expressed in the cytoplasm and nucleus, whereas VLPs were observed in the nucleus [12,30]. In this study, the interesting observation was that the NGPV VP2 protein was primarily localized in the nucleus (Figure 1A) and VLPs appeared in the nucleus. It was clear that VP2 assembled VLPs in the nucleus, and the efficient transportation of VP2 into the nucleus could accelerate the assembly of VLPs. The subunit vaccine was prepared using purified VLPs and used to immunize ducks. None of the immunized ducks exhibited adverse reactions, demonstrating that the VLP vaccine was safe for ducks.

In the challenge test, no deaths or clinical symptoms were observed in the immunized or unimmunized ducks. This result is inconsistent with that of Xiao et al., in which unimmunized ducks showed signs of slow growth and short beaks [30]. The possible reason for this phenomenon was that the ducks were challenged twice at the ages of 2 and 8 days in the work of Xiao et al.; however, the ducks were about 1 month old at the time of challenge in this study due to having two immunizations performed to produce a better immune effect. Since the ducks were susceptible to NGPV at 1–3 weeks old, NGPV infection in 1-month-old ducks showed low efficiency leading to no clinical symptoms. To solve this problem, increasing the challenge dose or reducing immune frequency should be performed to maintain the duck’s sensitivity to the virus in subsequent studies. Nevertheless, the virus could be detected in tissues and cloacal swabs, and the histopathological lesions were observed in livers and spleens after the challenge, suggesting that NGPV can infect 1-month-old ducks and the challenge models could be used to evaluate vaccine immunogenicity.

## 5. Conclusions

In this study, VP2 proteins were expressed and assembled into VLPs in the nuclei of Sf9 cells using a BEVS. The VLPs stimulated ducks to produce effective immune protection against NGPV, demonstrating that it can be used as a candidate vaccine for the prevention of NGPV.

## Figures and Tables

**Figure 1 vaccines-11-01768-f001:**
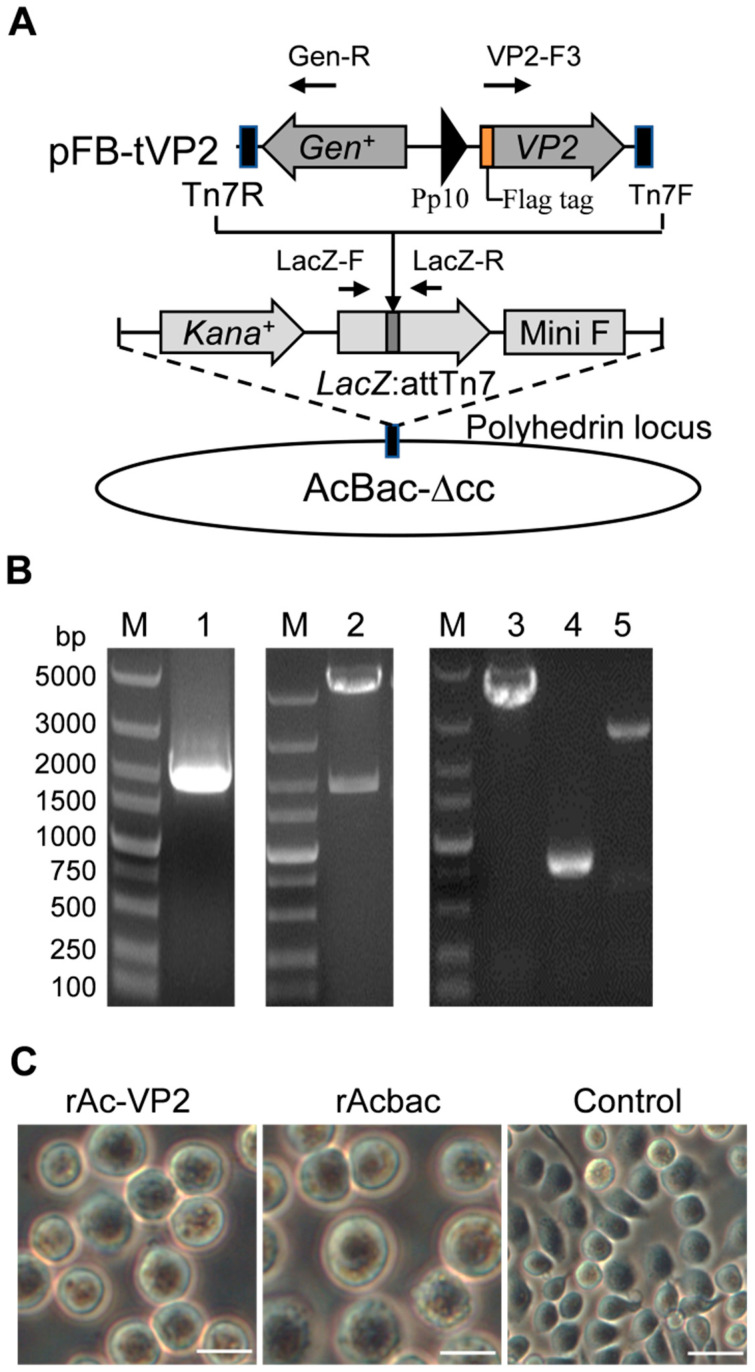
Construction of recombinant baculovirus genome and virus rescue. (**A**) The strategies for recombinant bacmid construction. A donor plasmid pFB-tVP2 is constructed by inserting the target gene tVP2 into pFastBac-dual plasmid, and then the tVP2 gene is converted to the bacmid AcBac-∆cc at the attTn7 site through transposition. The locations of primers, used for identifying the recombinant bacmid, are marked by horizontal arrows. (**B**) Amplification of target gene and the identification of donor plasmid and recombinant bacmid. M, DNA marker; line 1, polymerase chain reaction (PCR) amplification of target gene tVP2; line 2, the identification of donor plasmid pFB-tVP2 by digesting with restriction endonuclease *Xho* I + *Sph* I; lines 3–5, the identification of recombinant bacmid rAc-tVP2 by PCR using primer pairs of LacZ-F and LacZ-R, LacZ-F and Gen-R, and LacZ-R and VP2-F3. (**C**) Observation of pathological changes in *Spodoptera frugiperda* (Sf9) cells infected with recombinant baculovirus. The cultural supernatants of Sf9 cells, transfected with pAc-tVP2 or pAcBac-∆cc bacmid, are harvested at 120 h post-transfection and used to infect fresh Sf9 cells, and then the imagens are acquired at 96 h post-infection. Bar = 20 μm.

**Figure 2 vaccines-11-01768-f002:**
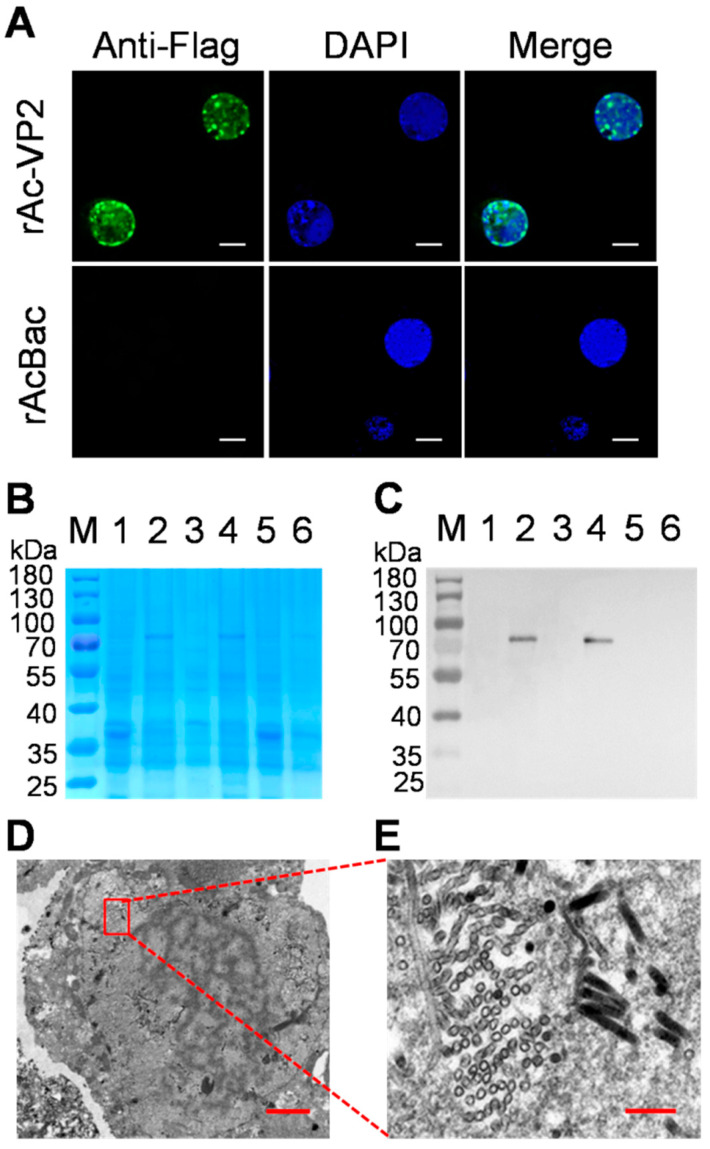
The authentication of tVP2 protein expression and the ultrastructure observation of the Sf9 cells infected by rAc-tVP2. (**A**) The identification of tVP2 protein via indirect immunofluorescence assay (IFA). The Sf9 cells are infected by rAc-tVP2 or rAcBac for 72 h, and the IFA program (green) is executed to detect the expression of tVP2 and the nucleus was stained with DAPI (blue) in cells. The bar = 20 μm. The identification of tVP2 protein via SDS-PAGE (**B**) and western blot (**C**). Line 1, line 3, and line 5 were the whole-cell, supernatant, and precipitation samples of the lysates of Sf9 cells infected by rAcBac, respectively; line 2, line 4, and line 6 contain the whole-cell, supernatant, and precipitation samples, respectively, of the lysates of Sf9 cells infected by rAc-tVP2, (**D**,**E**). The ultrastructure observation of the Sf9 cells infected by rAc-tVP2. The cells are infected with rAcBac-VP2 for 72 h, and the TEM procedure is conducted. The image in E is a higher magnification of the region in the red box in image D. Round or rod-shaped particles were found to aggregate in the nucleus. Bar = 2 μm and 200 nm for images (**D**,**E**), respectively.

**Figure 3 vaccines-11-01768-f003:**
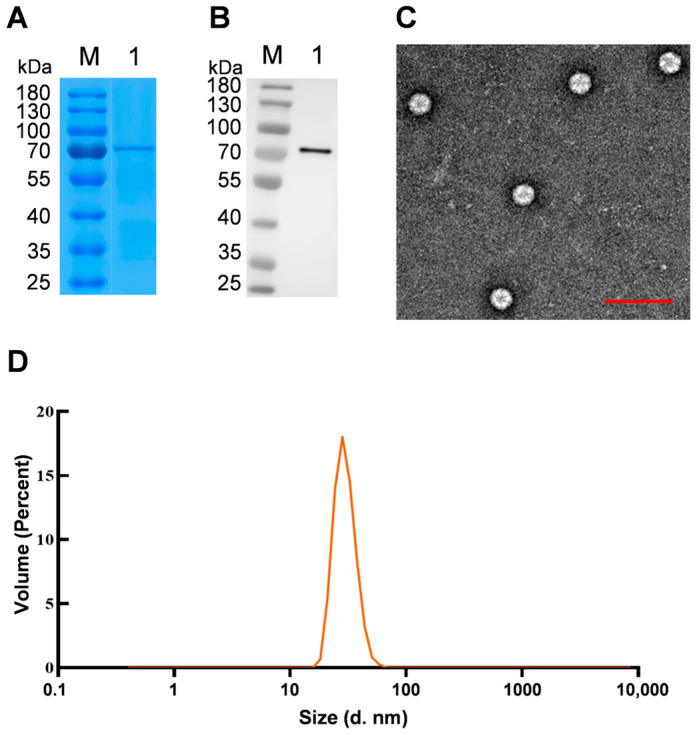
Purification and validation of virus-like particles (VLPs). (**A**,**B**) The validation of VLPs by sodium dodecyl sulfate polyacrylamide gel electrophoresis (SDS-PAGE) and western blot. The Sf9 cells, infected by rAcBac-VP2 for 96 h, are lysed and executed during the purification procedure, and then purified VLPs sample is detected via SDS-PAGE (**A**) and western blot (**B**). Line 1, the sample of purified VLPs. (**C**) Ultrastructure detection of purified VLPs. Bar = 100 nm. (**D**) Determination of particle size distribution via Zetasizer Nano ZS90 (Malvern Panalytical).

**Figure 4 vaccines-11-01768-f004:**
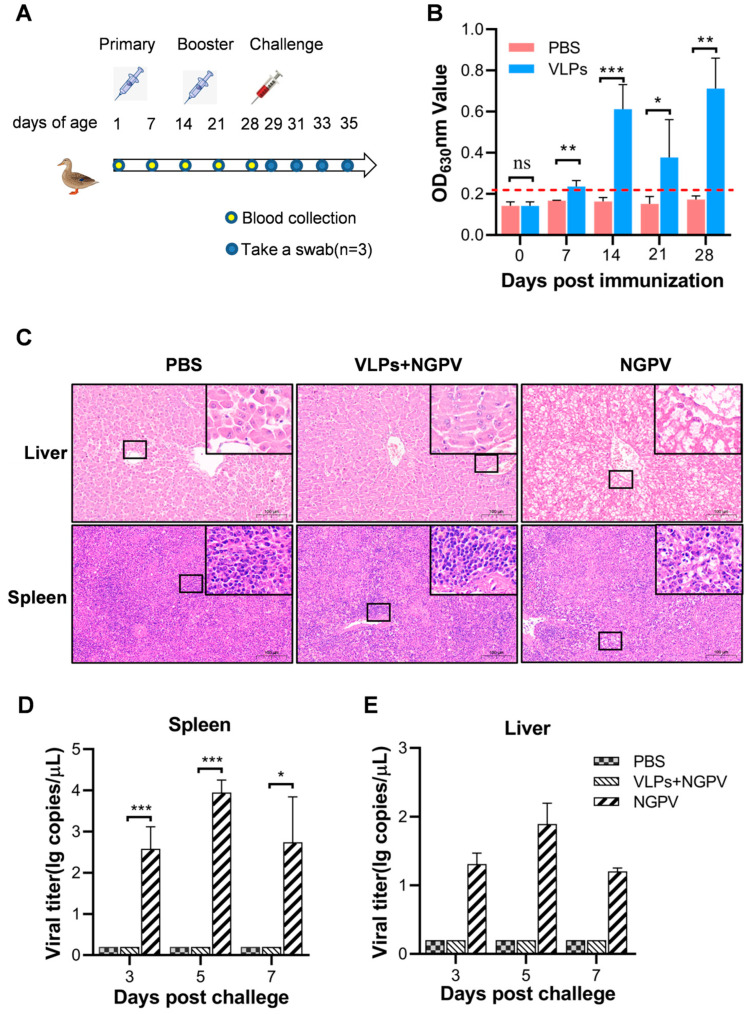
Immunization and challenge assays. (**A**) The flow chart of the animal procedures. The primary and booster immunization procedures are performed on ducks using virus-like particle (VLP) vaccines when aged 1 day old and 14 days old, respectively. The challenge assays are conducted at 14 days post-booster immunization. (**B**) Detection of antibody levels post-immunization. At 0, 7, 14, 21, and 28 days post-primary immunization. Blood samples are collected from ducks in the immune group (VLPs) and the control group (phosphate-buffered saline), and the sera are isolated. The enzyme-linked immunosorbent assay is used to detect the antibodies against the novel goose parvovirus. * *p* < 0.05, ** *p* < 0.01, *** *p* < 0.001. (**C**) Histopathological analysis of the liver and spleen. The liver and spleen tissues of ducks are collected at 5 days post-challenge (dpc) and examined using pathological analysis. Bar = 100 µm. (**D**,**E**) Detection of tissue viral load. The liver (**D**) and spleen (**E**) tissues of ducks are collected at 5 dpc and the tissue viral load is detected using quantitative real-time polymerase chain reaction. * *p* < 0.05, ** *p* < 0.01, *** *p* < 0.001.

**Table 1 vaccines-11-01768-t001:** Primers for construction and identification of recombinant baculovirus.

Name	Primer Sequences (5’–3’)
FH-F1	TTCTCGAGGCCACCATGGACTATAAGGACGACGATGATTACCACCATCACCATCATCAT (*Xho* I)
FH-R1	AGCTTCCCTGTATTTTTTTTTGCAGGTGCCGTATGATGATGGTGATGGTGGTAATCATC
VP2-F2	CCTGCAAAAAAAAATACAGGGAAGCT
VP2-F3	TTCTCGAGGCCACCATGGACTATAA (*Xho* I)
VP2-R2	TTGCATGCTTACAGATTTTGAGTTAGATATCTGGTTCCAAT (*Sph* I)
LacZ-F	TCACACAGGAAACAGCTATGACCATG
LacZ-R	CTCTTCGCTATTACGCCAGCTGG
Gen-R	TGCTGCCTTCGACCAAGAAGC

Note: The underlined parts represent the restriction sites of *Xho* I and *Sph* I.

**Table 2 vaccines-11-01768-t002:** Virus detection in cloacal swabs of vaccinated birds challenged with NGPV Yich strain at the indicated time points.

Group	Cloacal Swabs			
	1 dpc	3 dpc	5 dpc	7 dpc
PBS	0/10	0/10	0/10	0/10
VLPs + NGPV	0/10	0/10	0/10	0/10
NGPV	0/10	2/10	4/10	2/10

## Data Availability

Data are contained within the article.

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
