# Peer review of "Virus-Like Particles Based on the Novel Goose Parvovirus (NGPV) VP2 Protein Protect Ducks against NGPV Challenge"

_vaccines, 2023, doi:10.3390/vaccines11121768_

Round 1

Reviewer 1 Report

Comments and Suggestions for Authors

In this manuscript, the authors express the VP2 protein of the NGPV in Sf9 cells using baculovirus as vector. The recombinant VP2 protein was correctly expressed in the nucleus of Sf9 cells and formed VLPs that could be visualized by TEM. With these purified VLPs, oil formulations were made that were inoculated intramuscularly in 1 and 14 day old ducks. Two weeks after the booster dose, the animals were challenged with NGPV Yich strain intravenously. Although no clinical signs were observed in any group, the animals that received VLPs developed antibody response against NGPV and showed less histological damage in the liver and spleen.

Comments to the authors:

1) What is the production performance of VLPs? Is its large-scale production and purification feasible?

2) Why did the authors use an immunization protocol where the animals are less sensitive at the time of challenge?

3) Why is the challenge with NGPV Yich strain performed intravenously?

4) Is the challenge dose used normal for this model? Because in Chen et al.,2016; they use a higher dose. Considering the age at which animals are challenged in the protocol used in this manuscript, it would be advisable to try a higher dose.

5) Would it be possible to increase the dose of VLPs for a single dose and perform the challenge at a time when the animals are sensitive to the virus?

Minor comments:

Page 2, line 71: It says “virus, plasmid vector, cells and antibodies”, but no antibodies are indicated.

Author Response

Comments to the authors:

1) What is the production performance of VLPs? Is its large-scale production and purification feasible?

Reply:The yield of VLPs is 100-120 mg per 1L Sf9 cell culture, and 40-50 μg VLPs is need for one vaccination dose. The large-scale production and purification is achievable.

2) Why did the authors use an immunization protocol where the animals are less sensitive at the time of challenge?

Reply:In order to achieve the best immune effect, two immunizations were designed (refer to the research of Chen et al., 2012; DOI: 10.1016/j.virusres. 2012.08.009), however this resulting in the ducks being about 1 month old at the time of challenge. Although there were no clinical symptoms, the results of tissue viral load, pathological damage and virus shedding indicated that the virus was capable of infecting ducks at 1 month old.

3) Why is the challenge with NGPV Yich strain performed intravenously?

Reply:Thank you for reminding me. The method of challenge in this study was intramuscular injection (refer to the research of Chen et al., 2021; DOI: 10.1111/tbed.14021), and there was a clerical error in the introduction of the method. We have searched and modified the whole manuscript

4) Is the challenge dose used normal for this model? Because in Chen et al.,2016; they use a higher dose. Considering the age at which animals are challenged in the protocol used in this manuscript, it would be advisable to try a higher dose.

Reply:In previous studies, the challenge dose of 104.0 EID50 could cause significant clinical symptoms in 7 day-old ducklings, but there were hardly any clinical symptoms for 28 day-old ducks in this research. So it is necessary to further study the sensitivity of ducks in different days of age to the virus. However, the results of histopathological changes, viral load and virus shedding test suggested that ducks immunized with VLPs vaccine could reduce tissue viral load, histopathological damage and virus shedding after challenge with NGPV Yich strain, indicating that vaccine immunization could provide a better protective effect.

5) Would it be possible to increase the dose of VLPs for a single dose and perform the challenge at a time when the animals are sensitive to the virus?

Reply:To achieve the best immune effect, double immunization is often performed. It is good advice that the challenge is performed after one time immunization to maintain the animal's sensitivity to the virus. We will try this procedure in future studies.

Minor comments:

Page 2, line 71: It says “virus, plasmid vector, cells and antibodies”, but no antibodies are indicated.

Reply:The corresponding content in the manuscript has been corrected

Reviewer 2 Report

Comments and Suggestions for Authors

The authors describe the production and application of virus-like particles (VLP) in the baculovirus insect cell expression system. VLP of novel goose parvovirus (NGPV) were assembled from envelope protein VP2 and subsequently evaluated for their immunogenicity. After purification and detailed characterization, VLP were tested in an animal model including challenging experiments with virulent wild-type virus. VLP conferred protection and reduced viral load in tissue at several days post-challenge. NGPV VLP are suggested as a potential candidate for a vaccine against NGPV.

The report is a successful example of baculovirus produced VLP and their application as vaccine in an animal model. The manuscript is well written and concise and the results are organized systematically. Procedures are sufficiently explained except for one part which lacks essential information: the VLP purification procedure as outlined in the material and method section is extremely elaborate. A graphical illustration or flow chart would be very helpful to follow the individual steps. Separation of VLP from baculovirus infected insect cells is critical in general, especially when enveloped VLP are produced. They are similar to baculovirus particles in terms of surface properties since both derive their envelope from the cellular surface. Purification of non-enveloped VLP is also demanding and usually includes chromatography often in combination with gradient ultracentrifugation and filtration. I am wondering how the final purity of the vaccination material was assessed? Immunoblot analysis is shown for VLP, why not for gp64 (e.g. AcV5-antibody) to detect potential contamination. In my opinion such a verification is obligatory prior to immunization experiments. Considering the efficiency of the purification, what is the material balance and yield of this process (e.g. how many mL Sf9 culture are required for one vaccination dose) ?

Minor:

line 233: “…LacZ-F and VP2-F3.” should read “…LacZ-R and VP2-F3.”

line 373: “…symptoms occurred.” should read “…symptoms.”

Author Response

Comments to the authors:

The report is a successful example of baculovirus produced VLP and their application as vaccine in an animal model. The manuscript is well written and concise and the results are organized systematically. Procedures are sufficiently explained except for one part which lacks essential information: the VLP purification procedure as outlined in the material and method section is extremely elaborate. A graphical illustration or flow chart would be very helpful to follow the individual steps. Separation of VLP from baculovirus infected insect cells is critical in general, especially when enveloped VLP are produced. They are similar to baculovirus particles in terms of surface properties since both derive their envelope from the cellular surface. Purification of non-enveloped VLP is also demanding and usually includes chromatography often in combination with gradient ultracentrifugation and filtration. I am wondering how the final purity of the vaccination material was assessed? Immunoblot analysis is shown for VLP, why not for gp64 (e.g. AcV5-antibody) to detect potential contamination. In my opinion such a verification is obligatory prior to immunization experiments. Considering the efficiency of the purification, what is the material balance and yield of this process (e.g. how many mL Sf9 culture are required for one vaccination dose)?

Reply:(1) SDS-PAGE and WB were used to analyze the purity of the protein. The purified VLPs contained only one protein and could react with specific antibodies, indicating high purity of the VLPs. (2)The results of particle size test showed that the particles in the purified VLP were homogeneous and about 30nm in size which were similar to wild-type parvovirus, but significantly different from baculovirus particles or nuclear capsid. (3) The results of transmission electron microscopy further confirmed that the purified VLPs particles were about 30nm in size with round or hexagonal shape, which were similar to the morphology of parvovirus particles. And no baculovirus particles or nuclear capsid were found under transmission electron microscopy. These results indicated that we had obtained high purity VLPs samples.

   GP64 is a membrane protein of baculovirus, located on the viral envelope or cell membrane. The assembly of parvovirus VLPs is located in the nucleus. In this research, free baculovirus particles in the medium were removed easily by centrifugation when infected cells were collected; GP64 on the cell membrane could be easily removed by centrifugation after the collected cells were lysed. Therefore, GP64 protein was almost absent in the purified VLPs samples.

   The yield of VLPs is 100-120 mg per 1L Sf9 cell culture, and 40-50 μg VLPs is need for one vaccination dose.

Minor:

line 233: “…LacZ-F and VP2-F3.” should read “…LacZ-R and VP2-F3.”

Reply:The corresponding content in the manuscript has been corrected

line 373: “…symptoms occurred.” should read “…symptoms.”

Reply:The corresponding content in the manuscript has been corrected

Round 2

Reviewer 1 Report

Comments and Suggestions for Authors

I thank the authors for their responses.

I would just like to suggest that the authors consider adding some of the answers they gave me, in the Discussion section, to enrich the discussion.

Author Response

Comments and Suggestions for Authors: (1) I would just like to suggest that the authors consider adding some of the answers they gave me, in the Discussion section, to enrich the discussion. Reply: The discussion sections of the manuscript have been amended